# A Novel Quantitative Prediction Approach for Pungency Level of Chinese Liquor (Baijiu) Based on Infrared Thermal Imager

**DOI:** 10.3390/foods10051107

**Published:** 2021-05-17

**Authors:** Yingxia He, Shuang Chen, Ke Tang, Yan Xu, Xiaowei Yu

**Affiliations:** 1Key Laboratory of Industrial Biotechnology of Ministry of Education, Laboratory of Brewing Microbiology and Applied Enzymology, School of Biotechnology, Jiangnan University, Wuxi 214100, China; 7170201008@stu.jiangnan.edu.cn (Y.H.); shuangchen@jiangnan.edu.cn (S.C.); tandy81@jiangnan.edu.cn (K.T.); 2State Key Laboratory of Food Science & Technology, School of Biotechnology, Jiangnan University, Wuxi 214100, China

**Keywords:** pungency, infrared thermal imager, tongue surface temperature, Baijiu

## Abstract

Pungency is a crucial sensory feature that influences consumers’ appreciation and preferences toward alcoholic beverages. However, the quantitation of pungency is challenging to achieve using sensory analysis because of persistence, accumulation, and desensitization to the pungency perception. This study aimed to design a novel pungency evaluation method based on the measurement of tongue surface temperature. An infrared thermal (IRT) imager technique for measuring tongue surface temperature was established. To validate its feasibility, the IRT technique was used to measure tongue surface temperatures after the tongue was stimulated by (1) water and Baijiu, (2) different concentrations of ethanol aqueous solution (10, 20, 30, 40, and 50%, *v/v*), (3) ethanol aqueous solution and Baijiu samples with the same ethanol content, and (4) 26 Baijiu samples with different pungency level. For all cases, tongue surface temperatures showed large differences as a result of the different stimulation. The results showed that the tongue surface temperature correlated with the pungency intensity obtained by the sensory analysis. The relationship between tongue surface temperature and pungency intensity was established by multiple linear regression analysis. The IRT technique was able to be a useful support tool to quantitatively predict the pungency of alcoholic beverages, based on the measurement of tongue surface temperature.

## 1. Introduction

Alcoholic beverages are mainly consumed for hedonic pleasure, appreciated for the flavor and sensory quality [1]. Among the flavor, pungency sensation is an essential characteristic and is increasingly recognized to relate to sensory quality, palatability, and consumer preference of alcoholic beverages [2,3]. Currently, the evaluation of the pungency level in alcoholic beverages has been usually conducted by sensory analysis [4,5,6,7]. However, pungency is characterized by persistence, accumulation, and desensitization [8]. The sensory evaluation of pungency is generally perceived as a challenging task, owing to the low reproducibility, subjectivity, and taste fatigue of assessors [9]. This is notably the case for Chinese liquor (Baijiu), one of the largest distilled spirits in sales; its high level of ethanol content (38 to 65%, *v/v*) [10] may cause the possible desensitization on the tongue. Therefore, a novel method is needed to enable quantitative evaluation of pungency of alcoholic beverages.

Pungency results from the activation of the transient receptor vanilloid 1 (TRPV1) in the mouth by a variety of pungent substances to generate heat or pain sensation [11]. The capsaicin is known to activate heat receptor TRPV1 and induce changes in thermoregulatory processes with heat loss responses [12,13]. It is a common experience that an intake of pungent foods results in sweating and flushing of the face, which are the typical heat loss responses. According to heat transfer theory, skin temperature is influenced by various factors, including tissue thermal conductivity, the peripheral environment, metabolism, and physiological conditions [14]. Among these factors, blood perfusion is a critical factor, and the local skin surface temperature is directly related to subcutaneous blood circulation [15,16]. Boudreau et al. (2009) [17] reported that the topical capsaicin application on orofacial tissues could increase the flow of cutaneous blood and elevate the skin temperature. Nielsen and Gazerani (2013) [18] also demonstrated that capsaicin increases the cutaneous blood flow and skin temperature. Wu et al. (2017) [15] characterized the relationship between the skin blood flow and the skin temperature by a vasomotor response model. It is suggested that the stimulation of pungent substances can induce changes in skin temperature with the response of skin blood flow regulation. Lv et al. (2019) [19] also confirmed that tongue surface temperature can be affected by physical stimulation and chemical stimulation. An interesting idea arising is that the tongue surface temperature might be useful as an index for pungency evaluation.

The infrared thermography (IRT) is a non-invasive, quick, and reliable imaging technology for measuring surface temperature of any object [20]. IRT can visualize and record the temperature distribution of regions of interest with high resolution images. It has been widely used in the field of architecture [21,22], human medicine [23,24], and disease surveillance [16,25]. For example, recently, IRT has been used for mass fever screening at the entrance of subway, hospitals, airports, and border crossing points, to diagnose suspected visitors associated with COVID-19. IRT can be used as a powerful tool for the measurement of tongue surface temperature.

The aims of this study were: (i) to establish the IRT method for measuring tongue surface temperature, (ii) to validate the feasibility of the IRT method for measuring the changes of tongue surface temperature after different stimulation, and (iii) to characterize the relationship between tongue surface temperatures and pungency intensity of Baijiu. With the final goal to develop an accurate, fast, and convenient instrumental technique for quantitative prediction of pungency intensity of alcoholic beverages.

## 2. Materials and Methods

### 2.1. Materials and Samples

Bottled non-carbonated drinking water (550 mL each, Wahaha, Hangzhou, China) was purchased from a local supermarket. The different concentrations of aqueous ethanol solutions were prepared by diluting (98% ABV) ethanol (USP Grade, Merck, &Co. Shanghai, China) with mineral water. Twenty-six Baijiu samples with different pungency levels were provided by China Alcoholic Drinks Association. The details of the samples were given in Appendix A. All the samples were diluted to 46% ethanol content prior to test.

### 2.2. Establishing Infrared Thermal (IRT) Imaging Method for Tongue Surface Temperature Measurements

#### 2.2.1. Participants

A total of twelve participants (five females, seven males, aged between 21 and 31 years, body mass index of 19.2–23.8 kg/m^2^) were recruited among the postgraduate students of the School of Biotechnology at Jiangnan University. All participants are healthy and without medical complications related to their oral cavity. Before the tests, participants were provided with a written informed consent and instructed about the procedure of the test.

#### 2.2.2. Infrared Thermal (IRT) Imager

An infrared thermal imager (Testo 875-1i, Testo Instruments International Trading Co., Shanghai, China) has a scene range temperature of −20 °C to 100 °C, and the detection temperature was as small as 0.1 °C. It has a resolution of 160 × 120 pixels, a spectral range of 8–14 μm, and a lens of 32° × 23°. To evaluate the camera’s performance and accuracy prior to use, a calibration check was performed using water with known temperature values that was measured by a digital thermometer (DT) with a k-type thermocouple.

#### 2.2.3. IRT Image Acquisition

All tests were performed in a laboratory designated specifically for sensory analysis. The room temperature (25 ± 1 °C) was controlled by air conditioning and the humidity (50 ± 10%) was controlled by a humidifier. The laboratory entry was designed to prevent any interference. Proper and stable lighting in the laboratory room was maintained by lights and curtains, and ventilation was not allowed. In order to approximately cover the same number of pixels at each time during tests, the distance and angle of lens are fixed to minimize handling errors. During the test, a wooden support with visible mark was used to help all subjects precisely place their jaws on the same spot. IRT was placed ahead of the support using a fixed tripod. The lens and support were maintained to an angle of 30° and the lens was set with a distance of 0.20 m in the direction of 30° to tongue surface. The sketch of the infrared thermal images collection for tongue surface is presented in Figure 1. To ensure the optimum output stability of thermal camera, it was switched on one hour before use and the focus was manually adjusted to ensure high quality imaging.

Besides, we developed a standardized IRT test procedure for participants. Before the tests, subjects were asked not to exercise vigorously or eat for at least 1 h. In the laboratory, they were asked to relax for at least 15 min with no external disturbance. During the tests, subjects were instructed to clean their mouths with mineral water, take the sample (5 mL), and spread it all over the tongue surface. Then, they spit out the sample at the 5 s and protruded their tongues downward to the desired position immediately at the 15 s to capture infrared thermal images (a stopwatch was used for timing). To avoid the effect of test samples temperature on the results, all test samples were kept in a water bath at 30 °C, which is nearly the minimum tongue surface temperature. To avoid the effect of saliva and air movement on tongue surface, subjects were told not to respire through their mouth and to finish the saliva swallowing before protruding their tongues. The subjects were requested to wait at least 10 min between samples.

#### 2.2.4. IRT Image Processing

IRT cameras record thermal information and display it in the form of a color map (color palette). To enable the analysis of a thermal image visually, various color palettes can be used to assign the temperature readouts values to specific color. The captured thermographic images were analyzed by the dedicated software (Testo IRSoft, version 4.0) and the “Iron” palette was selected in our study. To obtain reproducible infrared temperatures, thermograms were adjusted for parameters of emissivity, environment temperature and humidity, and reflected temperature. Based on previous study, the emissivity value of the imager was set at 0.98 [19]. Since dimensions of the human tongue vary from one individual to another, in order to minimize the errors that resulted from tongue dimensions, tongue surface was divided into five areas and the areas were defined in proportion to the tongue size of the individual, as shown in Figure 2. The previous study reported that temperature sensitivity of four areas on tongue surface are in the following order: middle area > right lateral ≈ left lateral > tongue tip [19]. Thus, the whole area (ROI 1) and the middle area (ROI 2) of tongue surface were selected as the region of interest (ROI) in our study. Using one of the subjects as an example, a polygon tool was applied on the ROI 1 and a rectangle tool was applied on the ROI 2 (Figure 2). ROIs were used to extract temperature parameters to calculate the temperature index.

### 2.3. Validating the Feasibility of Infrared Thermal (IRT) Imaging Method for Tongue Surface Temperature Measurements

#### 2.3.1. Validating the Stability of Infrared Thermal Imaging (IRT) Method

In order to ensure the stability and accuracy of the established IRT method for tongue surface temperature measurement, a total of 12 subjects participated in the test. Their tongue surface temperatures after rinsing with the same Baijiu sample were recorded at the same time for three consecutive days. For each tongue image, the maximal temperature (T Max), the minimal temperature (T Min), and the average temperature (T Aver) were calculated for the regions of interest of ROI 1 and ROI 2 (Figure 2).

#### 2.3.2. Stimuli Application

Subjects were tested with different solutions and samples during three parts. In the first part, 12 subjects were treated with mineral water and the Baijiu sample. In the second part, the different concentrations of aqueous ethanol solutions were used as stimulus of different pungency level. The three selected subjects were treated with aqueous ethanol solutions of 10%, 20%, 30%, 40%, and 50% within random order. In the third part, the three selected subjects were treated with the 46% aqueous ethanol solution and two Baijiu samples (46%, *v/v*) in a random order.

### 2.4. Application of Infrared Thermal (IRT) Imaging Method for Quantitative Predicting Baijiu Pungency

#### 2.4.1. Tongue Surface Temperatures after Baijiu Application

The three selected subjects were treated with 26 Baijiu samples. For each tongue image, the maximal temperature (T Max), the minimal temperature (T Min), and the average temperature (T Aver) were calculated for the regions of interest (ROI 1 and ROI 2).

#### 2.4.2. Time-Intensity (TI) Analysis for Pungency Intensity

A trained panel of 12 assessors (7 males and 5 females, aged 21–28) [26] conducted the TI analysis of pungency. The assessors who all had previous TI analysis experience were further trained for pungency intensity evaluation based on the Standard Guide for Time-Intensity Evaluation of Sensory Attributes requirements [27]. They were trained to rate the pungency intensity on a 10-cm line scale from “0 = not present” on the left end of scale to “10 = extreme” on the right end of scale. The different concentrations of aqueous ethanol solutions were used as references of the scale for different pungency level. The scale was marked with 10%, 30%, and 50% aqueous ethanol solutions at 1.50 cm (slight pungency), 5.00 cm (moderate pungency), and 7.00 cm (strong pungency), respectively. They practiced continuously rating pungency intensity with at least 12 simulated TI analyses. The performance of the assessors was assessed by evaluating replicate TI curves for pungency. The training was considered sufficient if the curves were aligned at least 40% of the time [28].

After training sessions, the evaluation sessions were performed in isolated, temperature controlled (22 °C) tasting booths. The samples were served in coded tasting cups and presented in a randomized order. All of the samples were evaluated in duplicate. To avoid the possible desensitization, the assessors attended one session per day and the number of tasting samples were limited to three per session. For further recovery, the assessors were requested to wait at least 10 min between samples. Skim milk was available for assessors to use as a palate cleanser. To mask any interfering odors, all of the assessors wore nose-clips during their evaluations.

The assessors were instructed to sip the sample (2 mL), start the evaluation, manipulate the sample in their mouths for 5 s, swallow, and continue to rate the pungency intensity until they no longer perceived pungency, at which point they ended the recording. Rating was performed using a computer and mouse on a 10 cm line scale from “0 = not present” on the left end of scale to “10 = extreme” on the right end of the scale. The SensoMaker [29] was used as the interface for data acquisition.

### 2.5. Data Analysis

The maximal temperature (T Max), the minimal temperature (T Min), and the average temperature (T Aver) of regions of interest (ROI 1 and ROI 2) were calculated using professional software (Testo IRSoft, version 4.0). All statistical analyses were performed using SPSS 23.0 statistical software. The significant differences in the temperatures among different treatments were assessed by analysis of variance (ANOVA). For all statistical analyses, *p* < 0.05 was considered to be significant. The area under the curve (AUC) of the time-intensity curve for pungency evaluation was obtained for each repetition (12 evaluations per repetition) and each Baijiu sample by SensoMarker software (version 1.92, France) [29]. Correlations and the fitting equation between pungency intensity and tongue surface temperatures were established using XLSTAT software (version 2014; Add in soft, Paris, France).

## 3. Results and Discussion

### 3.1. Validating the Stability of the Infrared Thermal (IRT) Imaging Method

The tongue surface temperature measurements are sensitive and could be influenced easily by several factors [30,31,32]. Schaefer et al. (2012) [33] emphasized that the standardization of the method (angle and length) is necessary to take the thermographic picture. McCafferty (2007) [34] and Church et al. (2014) [35] demonstrated that ambient factors could influence the validity of results. To address these factors and achieve an accurate infrared thermal image, the IRT method for tongue surface temperature was established by us. It includes the setup of test room (ambient temperature, humidity, air circulation, lighting, etc.), subject control (physically relaxed, the consistent test protocol, etc.), the instrument control (lens focus, distance to subjects, lens angle, etc.), and data analysis (definition of regions of interest, color palette, etc.). A standardized framework was developed to perform reproducible measurement of temperatures of tongue surface.

As shown in Table 1, temperatures (T Min, T Max, and T Aver) of the tongue surface (ROI 1 and ROI 2) from 12 subjects after rinsing with the same Baijiu were recorded for three consecutive days by established IRT method. The results showed that no significant difference was observed during the three consecutive days (*p* > 0.05) for all subjects. Integrating the information of tongue surface temperatures from 12 subjects (data not shown), it was found that the T Aver of ROI 2 is higher than in ROI 1. The T Aver of ROI 1 and ROI 2 were 34.38 °C (ranging from 33.7 to 35.3 °C) and 34.82 °C (ranging from 34.2 to 35.9 °C), respectively. Jiang and Zhu (2007) reported a mean tongue surface temperature of 33.55 °C (ranging from 32.7 to 34.3 °C) based on 20 healthy subjects. Lv et al. (2019) [19] reported that in a similar controlled environment, the average tongue surface temperature based on 10 healthy subjects was 34.14 °C (ranging from 33.2 to 35.7 °C). The results of Table 1 showed that the measurements of tongue surface temperatures of subjects were fairly repeatable, reflecting the reliability and stability of the IRT method we established.

### 3.2. Tongue Surface Temperatures after Stimuli Application

Subjects were treated with different solutions during three parts. In the first part, the tongue surface temperatures of 12 subjects were measured by the established IRT method after mouth rinsing with mineral water and Baijiu sample (63.5% ethanol content). As shown in Table 2, the Baijiu treatment caused significant increase of tongue surface temperatures (*p* < 0.05). The T Min, T Max, and T Aver of the tongue surface ROI 1 increased by 0.55, 1.06, and 0.77 °C, respectively. The T Min, T Max, and T Aver of the tongue surface ROI 2 increased by 0.66, 0.74, and 0.65 °C, respectively. The results confirmed that ethanol stimulation could cause increase of tongue surface temperature. Lv et al. (2019) [19] also found that the temperatures of tongue surface area are higher after capsaicin treatment than that of control. The variations of tongue surface temperature after stimuli treatment could be due to the changes of cutaneous blood flow [16]. Bouzida et al. (2009) [17] found that the application of capsaicin on orofacial tissues increases the blood flow and temperature, paralleled with intense burning pain. Nielsen and Gazerani (2013) [18] demonstrated that capsaicin increases the cutaneous blood flow and skin temperature. Besides, we found that different subjects have different tongue sensitivity to stimulus. With the final goal of the development of an accurate, fast, and convenient IRT method for measuring tongue surface temperature, three subjects with good response sensitivity and results consistency were selected for later testing.

In the second part, the three selected subjects were treated with different concentrations of aqueous ethanol solutions (10%, 20%, 30%, 40%, and 50%). As shown in Table 3, with increasing concentrations of aqueous ethanol solutions, the T Aver of the tongue surface ROI 1 and ROI 2 show a consistent increase. As the tongue was exposed to the open air for temperature measurement, a quicker volatilization was expected with a higher ethanol content, which may be one of the reasons that temperature increased not significantly.

In the third part, the tongue surface temperatures of three selected subjects were measured after mouth rinsing with 46% aqueous ethanol solution (46% YCS) and two Baijiu samples with different aging times (which were diluted to the 46% ethanol content). As shown in Table 4, the T Min, T Max, and the T Aver of the tongue surface ROI 1 and ROI 2 are in the following order: young Baijiu > old Baijiu > 46% YCS. Histograms of temperature distribution of the tongue surface ROI 1 from one representative subject after being treated with 46% YCS, old Baijiu, and young Baijiu are shown in Figure 3a–c, respectively. All thermograms are presented using the same temperature scale (32.0 °C−37.0 °C). When comparing the frequency curve of temperature distribution (Figure 3d), it showed that the differences of the tongue surface temperatures could be mainly due to the proportion of temperature points in different temperature ranges. For example, after 46% YCS treatment, the tongue surface temperature ranged from 32.4 °C to 35.8 °C, and the highest proportion of temperature points were in 33.8 °C–34.0 °C (Figure 3a). After young Baijiu treatment, the tongue surface temperature ranged from 33.4 °C to 37.0 °C, and the highest proportion of temperature points were in 35.6 °C–35.8 °C (Figure 3c). The above results indicate that the established IRT method has the ability to characterize the differences of tongue surface temperatures after treatment with different solutions and samples.

We also evaluated the pungency of 46% YCS and two Baijiu samples with different aging times (which were diluted to 46% ethanol content) using the 3-AFC method. The results from 24 panelists showed the pungency intensity in the following order: young Baijiu > old Baijiu > 46% YCS (α = 0.01), which is consistent with the results of tongue surface temperature. In addition, this result suggests that the pungency of Baijiu might be influenced by other factors and chemical compounds besides ethanol. In previous studies, the TRPV1 and TRPA1 were found to be activated by α, β-unsaturated dialdehydes and aldehydes, respectively, such as, isovelleral, 4-hydroxynonenal, acetaldehyde, acetal, and acrolein [36,37]. One recent study suggests that carbonyl compounds influence the trigeminal burn of ethanol via the activation of TRPV1 and TRPA1 receptors [2]. Diallyl disulfide, a kind of volatile sulfur-containing compound, has been identified in Baijiu [38], which is also known to be a compound of allium and can activate the TRPV1 and TRPA1 receptors [39]. These volatile compounds are suggested to contribute to the pungency sensation of Baijiu that likely activate or act synergistically (with ethanol) to activate TRP receptors.

### 3.3. Relationship between Tongue Surface Temperatures and Pungency Intensity

The aforementioned results show that tongue surface temperatures and pungency intensity perceived are correlated. Therefore, we aimed to establish the relationship between pungency intensity and tongue surface temperature. The three selected subjects were treated with 26 Baijiu samples and their tongue surface temperatures were measured by the IRT method (Appendix A). Time-intensity (TI) is the most common dynamic method for evaluating perception changes during consumption, which can provide a detailed development of attribute intensity over time by continuous measurement [40]. TI has been widely used for intensity evaluation of time dependent attributes, such as astringency [41], and pungency [42]. Thus, the pungency intensity of these 26 Baijiu samples was evaluated by the TI analysis and characterized by the area under the curve (AUC) (Appendix A, Appendix A).

Regression analysis is a method used to establish the functional equation between a dependent variable or response and the variables that influence the response [43]. The temperatures of tongue surface ROI 2 and the pungency intensity of 26 Baijiu samples were analyzed by multiple linear regression. In order to build and test model, a dataset of 26 Baijiu samples was randomly separated into a training set of 22 Baijiu samples, which was used to build the model, and a test set of 4 Baijiu samples, which was then used to evaluate the built model. In a good regression model, a correlation should not exist between the independent variables, nor should it have multicollinearity [44]. With the selected variables, we have built the multiple linear regression model using the training set data and have obtained the following equation:Pungency = 133.19 Aver − 9.28 T Max − 15.10 T Min − 3362.84(1)

The obtained model was evaluated by statistical parameters such as squared multiple correlation coefficient (R^2^), adjusted correlation coefficient (R^2^adj), Fisher ratio (F), root mean-squared error (RMSE), Durbin–Watson statistic (DW), and significance (Sig). As it can be observed in Appendix A, N = 22, R^2^ = 0.648, R^2^adj = 0.590, RMSE = 31.506, DW = 2.043, F = 11.065, Sig = 0.000. The results show that the relationship between tongue surface temperatures and pungency intensity is statistically meaningful and significant. As shown in Figure 4, there was a good agreement between the simulated results and the experimental results. The results of the linear regression model suggest the potential of IRT as a novel technique in predicting the pungency difference of alcoholic beverages based on the measurement of tongue surface temperature.

In addition, the carbonation perception we experience upon consumption of carbonated beverages includes biting, tingling, stinging, pricking, and fizzy. It is well accepted that CO_2_ acts on neurons of the oral trigeminal nerve via a dual mechanism of action [45]. The presence of bubbles bursting in the mouth activates mechanoreceptors, whilst the CO_2_ converted via the carbonic anhydrase into carbonic acid that excites trigeminal neurons and elicits oral irritant sensations [46]. The increase of the CO_2_ level in water was found to be directly related to an increase in carbonation perception [47,48]. Therefore, the differences of the carbonation perception in carbonated beverages might also be quantitatively characterized based on the IRT method. Another interesting avenue for further research could be the investigation of the feasibility of IRT method to quantitatively measure the differences of carbonation sensations in carbonated beverages.

## 4. Conclusions

An infrared thermal (IRT) imager method for the measurement of tongue surface temperature was established in this study. The feasibility of the IRT method was validated, and the relationship between tongue surface temperature and pungency intensity was established by multiple linear regression analysis. The results suggest that the measurement of tongue surface temperature based on IRT technique could be a novel approach for quantitatively evaluating the pungency of alcoholic beverages, which can be used in sensory analysis and production practice.

## Figures and Tables

**Figure 1 foods-10-01107-f001:**
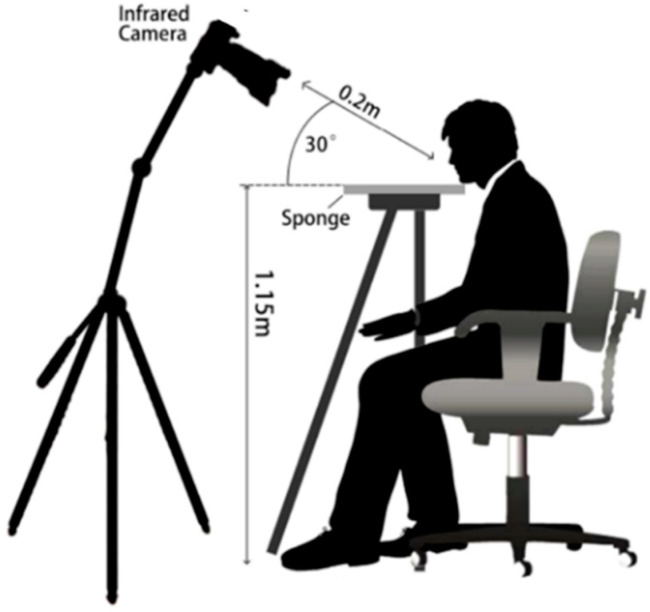
Sketch of thermography experiment for human subjects.

**Figure 2 foods-10-01107-f002:**
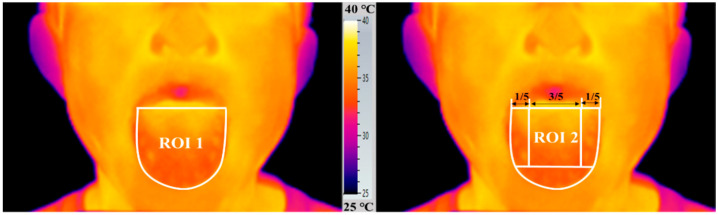
The regions of interest (ROIs) chosen for statistical analysis of the thermographic data. In each tongue image, the maximal temperature (T Max), the minimal temperature (T Min), and the average temperature (T Aver) were calculated for ROI 1 and ROI 2.

**Figure 3 foods-10-01107-f003:**
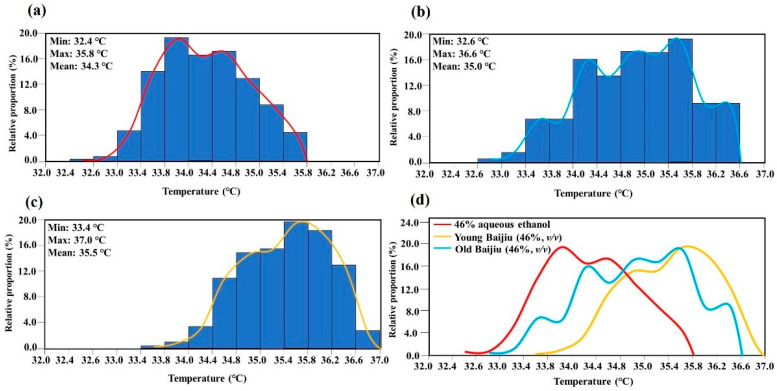
Temperature histogram of the tongue surface ROI 1 of one representative subject after being treated with different samples: (**a**) 46% aqueous ethanol, (**b**) old Baijiu with low pungency, (**c**) young Baijiu with high pungency, and (**d**) the frequency curve of temperature distribution. The vertical axis of the histogram represents the percentage of temperature points within the corresponding temperature range. The horizontal axis represents the range of temperature variations.

**Figure 4 foods-10-01107-f004:**
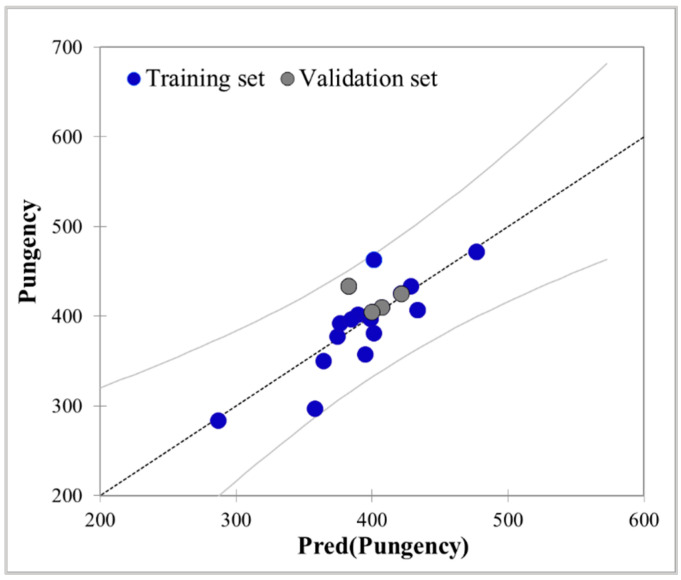
Comparison between predicted and observed values of the pungency calculated by the multiple linear regression model.

**Table 1 foods-10-01107-t001:** Temperature variations during three consecutive days of monitoring with IRT (*n* = 12).

Test Day	ROI 1			ROI 2		
	Min	Max	Aver	Min	Max	Aver
Day 1	32.60 ± 0.73	36.43 ± 0.42	34.70 ± 0.37	33.80 ± 0.65	36.16 ± 0.55	35.04 ± 0.53
Day 2	32.30 ± 0.57	36.27 ± 0.84	34.24 ± 0.72	33.47 ± 0.80	35.90 ± 0.75	34.70 ± 0.69
Day 3	32.11 ± 0.70	36.23 ± 0.72	34.21 ± 0.46	33.26 ± 0.59	35.99 ± 0.68	34.73 ± 0.48
*p* Value	0.311	0.803	0.120	0.259	0.712	0.132

**Table 2 foods-10-01107-t002:** Tongue surface temperatures after treated with mineral water and Baijiu (*n* = 12).

Sample	ROI 1			ROI 2		
	Min	Max	Aver	Min	Max	Aver
Water	31.86 ± 0.54	35.51 ± 0.76	33.77 ± 0.54	33.11 ± 0.48	35.39 ± 0.71	34.19 ± 0.55
Baijiu	32.41 ± 0.60	36.57 ± 0.58	34.54 ± 0.40	33.77 ± 0.54	36.13 ± 0.51	34.84 ± 0.34
*p* Value	0.094	0.013	0.011	0.033	0.043	0.02

**Table 3 foods-10-01107-t003:** Tongue surface temperatures after treated with aqueous ethanol solutions (*n* = 3).

Aqueous Ethanol Solutions	ROI 1			ROI 2		
	Min	Max	Aver	Min	Max	Aver
10%	28.87 ± 1.96	35.47 ± 1.01	32.53 ± 0.93	31.57 ± 0.65	34.87 ± 1.31	32.83 ± 0.90
20%	30.07 ± 1.19	35.80 ± 1.50	33.20 ± 1.50	32.13 ± 1.35	35.33 ± 1.08	33.47 ± 1.11
30%	30.23 ± 1.59	36.17 ± 1.32	33.53 ± 1.36	32.50 ± 1.18	35.67 ± 1.33	34.03 ± 1.15
40%	30.67 ± 0.45	35.47 ± 0.47	33.67 ± 0.91	32.03 ± 0.67	35.07 ± 0.15	34.27 ± 1.12
50%	31.37 ± 1.50	36.60 ± 1.25	34.33 ± 1.46	33.33 ± 1.53	36.17 ± 1.08	34.63 ± 1.33

**Table 4 foods-10-01107-t004:** Tongue surface temperatures after treated with the same ethanol content (46%, *v/v*) of ethanol aqueous solution and Baijiu samples (*n* = 3).

Samples	ROI 1			ROI 2		
	Min	Max	Aver	Min	Max	Aver
46% YCS	31.95 ± 0.94	35.85 ± 0.62	33.91 ± 0.66	33.00 ± 0.88	35.65 ± 0.48	34.33 ± 0.52
Old Baijiu	32.21 ± 0.58	36.21 ± 0.59	34.36 ± 0.50	33.26 ± 0.81	35.98 ± 0.30	34.71 ± 0.57
Young Baijiu	32.44 ± 0.74	36.38 ± 0.74	34.74 ± 0.55	33.51 ± 0.87	36.14 ± 0.60	35.04 ± 0.44
*p* Value	0.461	0.280	0.031	0.497	0.137	0.037

## Data Availability

The data that support the findings of this study are available from the corresponding author upon request.

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
