# Peer review of "A Novel Quantitative Prediction Approach for Pungency Level of Chinese Liquor (Baijiu) Based on Infrared Thermal Imager"

_foods, 2021, doi:10.3390/foods10051107_

Round 1

Reviewer 1 Report

In this manuscript, a simple infrared thermal imaging method for measuring the tongue surface temperature was established. The feasibility of this method was convincingly validated and the relationship between the tongue temperature and pungency intensity was esablished through multiple linear regression analysis.

  • Please clearly define the BMI as the Body Mass Index (in Mat & Meth)
  • Moreover, I would appreciate a discussion on the feasability of this method for quantitative measurement of the pungency of the carbonic bite (in carbonated alcoholic beverages, and carbonated soft drinks also). Actually, the presence of dissolved CO2 is also known to have a very characteristic tingling action in mouth. Could this also be correlated withe hte level of dissolved CO2 (because there is a wide range of dissolved CO2 in the carbonated drinks segment). I would appreciate a few words about this together with appropriate references.

Author Response

Response to Reviewer 1 Comments

In this manuscript, a simple infrared thermal imaging method for measuring the tongue surface temperature was established. The feasibility of this method was convincingly validated and the relationship between the tongue temperature and pungency intensity was established through multiple linear regression analysis.

Point 1: Please clearly define the BMI as the Body Mass Index (in Mat & Meth).

Response 1: Thanks for your comment and suggestion. We have changed the "BMI" to "Body Mass Index" in Material and Methods.

Point 2: I would appreciate a discussion on the feasability of this method for quantitative measurement of the pungency of the carbonic bite (in carbonated alcoholic beverages, and carbonated soft drinks also). Actually, the presence of dissolved CO2 is also known to have a very characteristic tingling action in mouth. Could this also be correlated with the level of dissolved CO2 (because there is a wide range of dissolved CO2 in the carbonated drinks segment). I would appreciate a few words about this together with appropriate references

Response 2: We very agree with your comments. Indeed, the carbonation perception (tingling, bite, stinging, pricking and fizzy) are very important for the overall quality perception, acceptability, and preference of carbonated beverages. It is well accepted that CO2 acts on oral trigeminal nerve via a dual mechanism of action. The presence of bubbles bursting in the mouth activates mechanoreceptors, whilst the conversion of CO2 to carbonic acid via carbonic anhydrase elicits a tingly response activating nociceptors [1,2]. The increase in the CO2 level was shown to be directly related to an increase in carbonation perception [3]. We have added discussion on the feasibility of IRT method for quantitative measurement the differences of carbonation perception of carbonated beverages, as following:

" In addition, the carbonation perception we experience upon consumption of carbonated beverages, including biting, tingling, stinging, pricking and fizzy. It is well accepted that CO2 acts on neurons of oral trigeminal nerve via a dual mechanism of action [1]. The presence of bubbles bursting in the mouth activates mechanoreceptors, whilst the CO2 converted via the carbonic anhydrase into carbonic acid that excites trigeminal neurons and elicits oral irritant sensations [2,4]. The increase of the CO2 level in water was found to be directly related to an increase in carbonation perception [3,5]. Therefore, the differences of the carbonation perception in carbonated beverages might also be quantitatively characterized based on IRT method. Another interesting avenue for further research could be the investigation of the feasibility of IRT method to quantitatively measure the differences of carbonation sensations in carbonated beverages. " (Lines 348-358)

References:

  1. Dunkel, A.; Hofmann, T. Carbonic anhydrase IV mediates the fizz of carbonated beverages. Angewandte Chemie 2010, 49, 2975-2977.
  2. Dessirier, J.M.; Simons, C.T.; Carstens, M.I.; O'Mahony, M.; Carstens, E. Psychophysical and neurobiological evidence that the oral sensation elicited by carbonated water is of chemogenic origin. Chemical senses 2000, 25, 277-284.
  3. McMahon, K.M.; Culver, C.; Castura, J.C.; Ross, C.F. Perception of carbonation in sparkling wines using descriptive analysis (DA) and temporal check-all-that-apply (TCATA). Food Quality and Preference 2017, 59, 14-26.
  4. Carstens, E.; Carstens, M.I.; Dessirier, J.M.; O'Mahony, M.; Simons, C.T.; Sudo, M.; Sudo, S. It hurts so good: oral irritation by spices and carbonated drinks and the underlying neural mechanisms. Food Quality and Preference 2002, 13, 431-443.
  5. McMahon, K.M.; Culver, C.; Ross, C.F. The production and consumer perception of sparkling wines of different carbonation levels. Journal of Wine Research 2017, 28, 123-134.

Reviewer 2 Report

The present manunscript from He and coworkers show an interesting account of thermal imaging applied to the prediction of pungency in Baijiu liquor. Overall, the methodology employed is well designed, although not well suited for reaching the goal stated in the manuscript title and introduction. In fact, also from the Authors’ own results, other factors than the alcohol content appear to clearly affect pungency (see comments ahead), although a certain correlation with ethanol could still be found. I would propose to the Authors to reconsider these aspects and accordingly adapt the manuscript's title and main target. Please, find here my specific comments.

1) Line 44 (and in the rest of the paper): The Authors targeted only ethanol as the cause for pungency in Baijiu. Accordingly, the panel was trained and the study carried on upon that assumption. Isn’t there any other known specific factor (e.g., volatile chemical compounds) that can elicit this receptor in this liquor, apart from ethanol? For example, diallyl disulfide (allicin) is a known compound in Baijiu (see for example: Song, X., et al. Food Chemistry, 2019, vol. 297, 124959), which is also known to be a compound of Allium activating the same TRPV1 receptor mentioned in the introduction (see for example: Macpherson, L.J., et al. Current Biology, vo. 15, issue 10, pp. 929-934). Therefore, the choice of calibrating pungency only against the concentrations of ethanol must either be justified, or at least clearly indicated in the abstract and the title of the manuscript.

2) Line 261-262: When comparing Table 1 and 2, the temperatures reached after testing the liquor are totally indistinguishable from the control ones measured at different days. How can the Authors prove that the measured temperature’s increases from water to Baijiu tasting is not just the result of a re-equilibration to normal tongue surface’s temperature? Did the Authors try also to record tongue temperatures without giving the panellists the sample of liquor after the water’s rinsing, instead giving them just water kept at the same temperature as the liquor?

3) Line 298: The Authors showed that “the results 298 from 24 panellists showed the pungency intensity are also in the following order: young 299 Baijiu > old Baijiu > 46% YCS (α=0.01)”. However, few lines before they indicated that they “also evaluated the pungency of 46% YCS and two Baijiu samples with different aging times (which were diluted to 46% ethanol content)”. So, all samples had the same ethanol content. May the Authors consider (and comment on) that, also according to their results, other factors might actually be affecting pungency in the liquor beside ethanol?

Author Response

Response to Reviewer 2 Comments

The present manuscript from He and coworkers show an interesting account of thermal imaging applied to the prediction of pungency in Baijiu liquor. Overall, the methodology employed is well designed, although not well suited for reaching the goal stated in the manuscript title and introduction. In fact, also from the Authors’ own results, other factors than the alcohol content appear to clearly affect pungency (see comments ahead), although a certain correlation with ethanol could still be found. I would propose to the Authors to reconsider these aspects and accordingly adapt the manuscript's title and main target. Please, find here my specific comments.

Point 1: Line 44 (and in the rest of the paper): The Authors targeted only ethanol as the cause for pungency in Baijiu. Accordingly, the panel was trained and the study carried on upon that assumption. Isn’t there any other known specific factor (e.g., volatile chemical compounds) that can elicit this receptor in this liquor, apart from ethanol? For example, diallyl disulfide (allicin) is a known compound in Baijiu (see for example: Song, X., et al. Food Chemistry, 2019, vol. 297, 124959), which is also known to be a compound of Allium activating the same TRPV1 receptor mentioned in the introduction (see for example: Macpherson, L.J., et al. Current Biology, vo. 15, issue 10, pp. 929-934). Therefore, the choice of calibrating pungency only against the concentrations of ethanol must either be justified, or at least clearly indicated in the abstract and the title of the manuscript.

Response 1: We very agree with your comments. Apart from ethanol, the pungency of Baijiu might be influenced by other factors and chemical compounds. In previous studies, the TRPV1 and TRPA1 were found to be activated by α, β-unsaturated dialdehydes and aldehydes respectively, such as, isovelleral, 4-hydroxynonenal, acetaldehyde, acetal, and acrolein [1,2]. One recent study suggests carbonyl compounds influence the burning perception of ethanol via the activation of TRPV1 and TRPA1 receptors [3]. Besides, as you mentioned in the comments, diallyl disulfide, a volatile sulfur-containing compound, has been identified in Baijiu [4], which is also known to be a compound of allium and can activate the TRPV1 receptor [5]. These volatile compounds are suggested to contribute to the pungency sensation of Baijiu that likely activate or act synergistically (with ethanol) to activate TRP receptors.

We are very sorry, maybe our description is unclear. As the pungency of Baijiu has been mainly attributed to ethanol content, in our study, the different concentrations of aqueous ethanol solutions were just used as references and scales of different pungency level for panel training and the validation of infrared thermal (IRT) imaging method. We have clearly indicated it the in the Materials and Methods of the revised manuscript, as following:

"In the second part, as the pungency of Baijiu has been mainly attributed to ethanol content, the different concentrations of aqueous ethanol solutions were used as stimulus of different pungency level. The three selected subjects were treated with aqueous ethanol solutions of 10%, 20%, 30%, 40%, and 50% within random order." (Lines 163-166)

"They were trained to rate the pungency intensity on a 10-cm linear scale from "0 = not present" on the left end of scale to "10 = extreme" on the right end of scale. As the pungency of Baijiu has been mainly attributed to ethanol content, the different concentrations of aqueous ethanol solutions were used as references of the scale for different pungency level." (Lines 180-184)

Point 2: Line 261-262: When comparing Table 1 and 2, the temperatures reached after testing the liquor are totally indistinguishable from the control ones measured at different days. How can the Authors prove that the measured temperature’s increases from water to Baijiu tasting is not just the result of a re-equilibration to normal tongue surface’s temperature? Did the Authors try also to record tongue temperatures without giving the panellists the sample of liquor after the water’s rinsing, instead giving them just water kept at the same temperature as the liquor?

Response 2: Thanks for the reviewer’s insightful and valuable comments. We are very sorry for our unclear description of Table 1. In order to validate the stability of the established infrared thermal (IRT) imaging method, tongue surface temperatures of 12 subjects after rinsing with a same Baijiu sample were measured during three consecutive days (Table 1). Table 2 shows the changes of the tongue surface temperatures after treated with a control solution (mineral water) and Baijiu sample. Therefore, there was no obvious difference in the tongue surface temperatures, when comparing Table 1 and Table 2.

We recorded the original tongue surface temperatures of subjects before treatment, as shown in Table S1. When comparing Table S1 with Table 1, there was no obvious differences in the tongue surface temperatures before treatment and after rinsing with water.

Table S1. The average tongue surface temperatures (ROI 1) of 12 subjects before treatment.

Test Day

Min

Max

Aver

Day 1

32.11

35.65

33.84

Day 2

31.61

35.42

33.44

Day 3

31.76

35.32

33.57

Point 3: Line 298: The Authors showed that “the results 298 from 24 panellists showed the pungency intensity are also in the following order: young 299 Baijiu > old Baijiu > 46% YCS (α=0.01)”. However, few lines before they indicated that they “also evaluated the pungency of 46% YCS and two Baijiu samples with different aging times (which were diluted to 46% ethanol content)”. So, all samples had the same ethanol content. May the Authors consider (and comment on) that, also according to their results, other factors might actually be affecting pungency in the liquor beside ethanol?

Response 3: We agree with the reviewer’s comments. Apart from ethanol, the pungency of Baijiu might be affected by other factors and chemical compounds. In our manuscript at lines 301-304, we have indicated this. We have supplied more discussion on other factors that affect the pungency sensation of Baijiu in the revised manuscript, as following:

" The results from 24 panelists showed the pungency intensity are in the following order: young Baijiu > old Baijiu > 46% YCS (α=0.01), which are consistent with the result of tongue surface temperature. In addition, this result suggests that the pungency of Baijiu might be influenced by other factors and chemical compounds besides ethanol. In previous studies, the TRPV1 and TRPA1 were found to be activated by α, β-unsaturated dialdehydes and aldehydes respectively, such as, isovelleral, 4-hydroxynonenal, acetaldehyde, acetal, and acrolein [1,2]. One recent study suggests that carbonyl compounds influence the trigeminal burn of ethanol via the activation of TRPV1 and TRPA1 receptors [3]. Diallyl disulfide, a kind of volatile sulfur-containing compound, has been identified in Baijiu [4], which is also known to be a compound of allium and can activate the TRPV1 and TRPA1 receptor [5]. These volatile compounds are suggested to contribute to the pungency sensation of Baijiu that likely activate or act synergistically (with ethanol) to activate TRP receptors. " (Lines 305-314)

References:

  1. Bang, S.; Kim, K.Y.; Yoo, S.; Kim, Y.G.; Hwang, S.W. Transient receptor potential A1 mediates acetaldehyde-evoked pain sensation. European Journal of Neuroscience 2007, 26, 2516-2523.
  2. Trevisani, M.; Siemens, J.; Materazzi, S.; Bautista, D.M.; Nassini, R.; Campi, B.; Imamachi, N.; Andre, E.; Patacchini, R.; Cottrell, G.S. 4-Hydroxynonenal, an endogenous aldehyde, causes pain and neurogenic inflammation through activation of the irritant receptor TRPA1. Proceedings of the National Academy of Sciences of the United States of America, 2007, 104, 13519-13524.
  3. Kokkinidou, S.; Peterson, D.G. Identification of compounds that contribute to trigeminal burn in aqueous ethanol solutions. Food chemistry 2016, 211, 757-762.
  4. Song, X.; Zhu, L.; Wang, X.; Zheng, F.; Zhao, M.; Liu, Y.; Li, H.; Zhang, F.; Zhang, Y.; Chen, F. Characterization of key aroma-active sulfur-containing compounds in Chinese Laobaigan Baijiu by gas chromatography-olfactometry and comprehensive two-dimensional gas chromatography coupled with sulfur chemiluminescence detection. Food chemistry 2019, 297, 124959.
  5. Macpherson, L.J.; Geierstanger, B.H.; Viswanath, V.; Bandell, M.; Eid, S.R.; Hwang, S.; Patapoutian, A. The pungency of garlic: activation of TRPA1 and TRPV1 in response to allicin. Current Biology, 2005, 15, 929-934.

Round 2

Reviewer 2 Report

Dear Authors,

I have read your replies and appreciated the effort of tackling the raised points. However, some crucial points should be sill be fully addressed. You replied:

1) 'We are very sorry, maybe our description is unclear. As the pungency of Baijiu has been mainly attributed to ethanol content, in our study, the different concentrations of aqueous ethanol solutions were just used as references and scales of different pungency level for panel training and the validation of infrared thermal (IRT) imaging method'.

Earlier instead you wrote:

2) 'Apart from ethanol, the pungency of Baijiu might be influenced by other factors and chemical compounds. In previous studies, the TRPV1 and TRPA1 were found to be activated by α, β-unsaturated dialdehydes and aldehydes respectively, such as, isovelleral, 4-hydroxynonenal, acetaldehyde, acetal, and acrolein [1,2]. One recent study suggests carbonyl compounds influence the burning perception of ethanol via the activation of TRPV1 and TRPA1 receptors [3]. Besides, as you mentioned in the comments, diallyl disulfide, a volatile sulfur-containing compound, has been identified in Baijiu [4], which is also known to be a compound of allium and can activate the TRPV1 receptor [5]. These volatile compounds are suggested to contribute to the pungency sensation of Baijiu that likely activate or act synergistically (with ethanol) to activate TRP receptors.'

So, how do you reconcile these two concepts? As pointed out in my revision, for particular samples (aged vs non-aged...) you might find pungengy levels totally uncorrelated with the ethanol content. So, instead of writing "As the pungency of Baijiu has been mainly attributed to ethanol content..." which is fundamentally flawed, could you stress (for instance) everywhere that you are using ethanol to model pungengy in general, although pungengy is not only due to ethanol in Baijiu?

Author Response

Response to Reviewer 2 Comments

I have read your replies and appreciated the effort of tackling the raised points. However, some crucial points should be sill be fully addressed. You replied:

1) 'We are very sorry, maybe our description is unclear. As the pungency of Baijiu has been mainly attributed to ethanol content, in our study, the different concentrations of aqueous ethanol solutions were just used as references and scales of different pungency level for panel training and the validation of infrared thermal (IRT) imaging method'.

Earlier instead you wrote:

2) 'Apart from ethanol, the pungency of Baijiu might be influenced by other factors and chemical compounds. In previous studies, the TRPV1 and TRPA1 were found to be activated by α, β-unsaturated dialdehydes and aldehydes respectively, such as, isovelleral, 4-hydroxynonenal, acetaldehyde, acetal, and acrolein [1,2]. One recent study suggests carbonyl compounds influence the burning perception of ethanol via the activation of TRPV1 and TRPA1 receptors [3]. Besides, as you mentioned in the comments, diallyl disulfide, a volatile sulfur-containing compound, has been identified in Baijiu [4], which is also known to be a compound of allium and can activate the TRPV1 receptor [5]. These volatile compounds are suggested to contribute to the pungency sensation of Baijiu that likely activate or act synergistically (with ethanol) to activate TRP receptors.'

So, how do you reconcile these two concepts? As pointed out in my revision, for particular samples (aged vs non-aged...) you might find pungengy levels totally uncorrelated with the ethanol content. So, instead of writing "As the pungency of Baijiu has been mainly attributed to ethanol content..." which is fundamentally flawed, could you stress (for instance) everywhere that you are using ethanol to model pungengy in general, although pungengy is not only due to ethanol in Baijiu?

Response: We very agree with your comments. Sincere appreciation for all of your thoughtful and rigorous suggestions. Apart from ethanol, the pungency of Baijiu might be influenced by other factors and chemical compounds. Both ethanol and other chemical compounds can activate the transient receptor vanilloid 1 (TRPV1) in the mouth to generate pungency sensation and induce changes in skin temperature with the response of skin blood flow regulation. In this study, we aimed to develop a novel approach to characterize the differences of pungency intensity based on the changes of tongue surface temperatures. We have revised the corresponding description in the revised manuscript according to your suggestion.